# What Can We Learn from Depth Camera Sensor Noise?

**DOI:** 10.3390/s22145448

**Published:** 2022-07-21

**Authors:** Azmi Haider, Hagit Hel-Or

**Affiliations:** Department of Computer Science, University of Haifa, Haifa 3498838, Israel; ahaide03@campus.haifa.ac.il

**Keywords:** depth camera, depth sensors, noise

## Abstract

Although camera and sensor noise are often disregarded, assumed negligible or dealt with in the context of denoising, in this paper we show that significant information can actually be deduced from camera noise about the captured scene and the objects within it. Specifically, we deal with depth cameras and their noise patterns. We show that from sensor noise alone, the object’s depth and location in the scene can be deduced. Sensor noise can indicate the source camera type, and within a camera type the specific device used to acquire the images. Furthermore, we show that noise distribution on surfaces provides information about the light direction within the scene as well as allows to distinguish between real and masked faces. Finally, we show that the size of depth shadows (missing depth data) is a function of the object’s distance from the background, its distance from the camera and the object’s size. Hence, can be used to authenticate objects location in the scene. This paper provides tools and insights into what can be learned from depth camera sensor noise.

## 1. Introduction

Depth cameras capture scene structure by evaluating the distance between points in the scene and the camera. Recent years have seen the release of consumer level depth cameras such as Microsoft Kinect depth cameras, Intel Realsense, StereoLabs ZED camera and others. These cameras have been used extensively for human body tracking, pose estimation, action recognition as well as for structure reconstruction, modeling, and many other uses. The capabilities, performance and especially the limits of these cameras have been studied and compared [1,2,3,4,5].

In this paper, we study depth cameras and specifically their noise properties. Although camera and sensor noise are often disregarded, assumed negligible or attempted to be removed in the context of denoising, in this paper we show that significant information can actually be deduced from depth camera noise about the captured scene and the objects within it. From sensor noise alone, object location in the scene can be deduce, consequently, 3D motion paths of can be constructed from noise alone. Sensor noise can indicate the source camera type, and even the specific camera unit that was used. Distribution of sensor noise on objects in the scene can indicate the light source direction and noise patterns can distinguish between real and masked individuals. Finally we show that missing depth data (shadows) in the scene impose a relationship between object distance from background, object distance from camera and object size. These insights and knowledge of the scene collected through the depth sensor noise have a direct implementation as indicators for depth image tampering. Thus, inconsistencies between measurements derived from the camera noise, and the actual values in the depth image, are strong indications that the image has been manipulated. Light source direction can provide estimates of the time of image capture and distinguishing between real and masked faces can assist in detection of spoofing attacks. Although a plethora of depth cameras are available, We restrict our analysis to consumer, low-cost and readily available cameras.

## 2. Background

### 2.1. Depth Imaging

Depth cameras use sensing technology to infer the distance (or depth) of points in the scene from the camera. They output image sequences in which each frame is a depth image where pixel values represent the distance from the camera (see Figure 1). Depth camera components include optics, sensors and the imaging pipeline [6]. Additionally these cameras incorporate components, unique to depth sensing such as IR projectors IR sensors, phase detectors and more.

Different technologies are used by depth cameras: (Figure 2):Stereo imaging [7,8]—Cameras using this technology consist of at least two cameras that capture standard 2D RGB images. They are positioned in parallel along a baseline and are activated simultaneously to capture multiple views of the same scene. Correspondences between points in the two views are determined and triangulation is used to compute the distance from camera of the 3D scene points.Structured light (Projected-light sensors) [9,10,11]—This active sensing technology, projects patterns of IR light onto the scene. The projected pattern of IR light is captured by the camera’s IR sensor. The patterns of light are designed to easily determine correspondences between the original light pattern that was projected and the pattern of light reflected from the scene. As in stereo imaging, triangulation is then used to compute the depth of the 3D scene points.Time of flight (ToF) [12,13]—Cameras with this sensing technology project an IR light wave which is reflected from objects in the scene and captured back by the camera’s IR sensor. The shift between the projected light and the captured reflected light allows to estimate the distance from the camera to the scene object. Two types of ToF cameras are used: Continuous wave modulation [14] in which the frequency of the projected IR wave is varied and the phase delay is measured to evaluate depth. Pulse light modulation [15] in which a very short pulse of light is projected and the time till its return is measured. This approach avoids dealing with phase and thus overcomes issues of phase ambiguity.

### 2.2. Depth Camera Sensor Noise

Similar to most acquisition systems, noise is also inherent in depth camera sensors and in the resulting depth-images. It arises from several sources. Camera build and technical specs such as focal length, field of view, quality of lenses, all affect image quality and consequently the noise in the resulting depth image. In active acquisition systems, the quality of the projected IR light, including its intensity and collimation, affect image noise (Figure 3). In ToF cameras, the quality of the IR signal modulation is a significant factor affecting noise levels. Camera build parameters such as the baseline distance between cameras for stereo based systems and the camera to projector distance in structured light based systems also strongly affect image depth quality [9,10,11,16,17].

Noise in depth images is strongly dependent on the method of depth acquisition. Stereo and structured light systems compute depth based on point correspondences across multiple views. Interpolation is performed between these points which introduces depth errors [9,10,11]. ToF methods based on phase are susceptible to phase ambiguity and demodulation errors [13] which produce incorrect depth estimates.

In addition to the noise introduced by the camera, scene layout and objects within the scene affect noise levels in depth images. Object color, brightness and material affect depth estimation [13,18] (Figure 4). This is most likely due to the variations in IR absorption across different materials [2]. At the extreme, specular surfaces and transparent objects introduce large depth errors [2,19]. The layout of objects within a scene can strongly affect depth noise, specifically depth evaluation of an object may be strongly affected by inter-reflection and light scatter from surrounding objects [18,20]. When object (or camera) motion occurs, motion blur in depth cameras result in overestimation or underestimation of depth near depth edges [21,22,23].

Finally, scene illumination is a major source of noise in depth images. Depth cameras do not perform well under strong illumination, specifically under natural outdoor lighting (see Figure 5). This is largely due to the IR components in natural light that tend to confuse the IR sensors of the depth camera. High intensity lighting may also introduce depth errors as it may dominate over the relatively low intensity IR light projected by the camera [5,24].

Noise in depth images show distinct characteristics and patterns. It increases along strong depth edges, due to difficulty in triangulation (passive stereo systems) or due to inconsistency in the reflected light (structured light and ToF systems) [2,19,25,26]. Additionally, IR light projection from the camera may be blocked by occluding objects within the scene, thus producing depth shadows in the image [2,19] (see Section 7).

Across the sequence of depth images produced by the depth-cameras, noise is expressed as inaccuracy and inconsistency of the estimated depth. It can be considered as either varying spatially or varying temporally at a pixel. *Axial noise* is the temporal fluctuation of depth values at a pixel across multiple frames (see Figure 6). It has been shown to increase quadratically with the distance of objects from the camera [2,3,4,18,27,28,29,30] (see Figure 7). The source of this behavior may be the decrease in amplitude of the projected IR light in ToF cameras or the dependency between disparity and depth in the stereo and structured light cameras.

Spatially varying depth inconsistencies, termed *Lateral Noise*, increases linearly from image center, in both the horizontal and vertical directions (see Figure 7). It tends to increase significantly at the edge of the depth image, possibly due to camera lens distortion [2,4,27,28,29,30]. Older cameras, show additional noise patterns such as a vertical stripe pattern or a radial ripple-like pattern of noise [2,4,27,30].

We define noise at a pixel as the deviation of depth values from the mean depth across a given number of frames. Noise magnitude is defined as the variance of the depth values and noise variance is defined as the variance of the absolute of deviation values:(1)$$noiseMagnitude=\frac{1}{K}\sum_{i}{\left(d_{i}-\bar{d}\right)}^{2}$$
(2)$$noiseVariance=var(|d_{i}-\bar{d}|)$$
where $d_{i}$ is the depth value in frame i and $\bar{d}$ the mean across *K* frames. Figure 6a plots the depth values at a pixel captured by a Kinect depth camera [31] across 50 frames. Figure 6b plots the histogram of deviations from the mean depth value. The noise magnitude is μ=0.094 cm, and noise variance is $\sigma^{2}=0.036$ cm.

## 3. Determining Object Position from Sensor Noise

In [32], sensor noise was exploited for forgery detection. The outcome of that study implies that position of objects in the scene can be determined from sensor noise alone. Sensor noise as defined in Equations (Equation 1) and (Equation 2), has a distinct pattern as a function of distance from the camera and as a function of deviation from the center of the camera’s field of view. Figure 7 shows noise magnitude as a function of depth (Z) and horizontal deviation (X) from the camera center as captured by a Kinect camera. A similar pattern is obtained for vertical deviation (Y) but for simplicity we restrict discussion to the horizontal direction. As can be seen, noise increases with depth as well as with horizontal deviation. This pattern allows to estimate the position of an object in the scene from the noise magnitude at the pixels associated with the object.

To verify this, a data set was collected by recording depth samples of an object placed at numerous positions in the scene using a KinectV2 [31]. The object was placed at 30 cm intervals symmetrically within the camera’s horizontal field of view (X position) and between 140 cm and 350 cm distance from the camera (Z position) resulting in 81 positions. For each sample, 300 frames were recorded, from which the noise magnitude and variance on the object were computed as a feature vector. The label of each sample was the ground truth x-z position from which the sample was recorded. A multi-class SVM classifier with squared hinge loss [33] was trained on the sampled feature vectors to form a division of the classes based on the sample noise. For testing, an additional 2025 samples were collected of the object placed again in the scene at the designated locations. Using the trained SVM classifier the position (X and Z values) were predicted and compared to the ground truth values. Table 1 shows the results. Accuracy of predicting the X-Z position of the samples showed 73% accuracy. Considering the top 2 rankings predicted by the classifier, improved accuracy to 92%. The low accuracy in the highest ranked prediction is explained in the confusion that occurs due to the symmetry in noise values along the horizontal (X) direction as can be seen both in Figure 7 and in the large average X error value in Table 1 compared to the average Z error. Similar experiments were performed for other camera types such as KinectV1 [34] (structured light), ZED [35] (stereo) and showed very similar behavior.

## 4. Determining Source Camera from Sensor Noise

In addition to variation in depth sensor noise as a function of object position, as shown above, it is also found to vary across camera types due to the make and the depth sensing technology used in the cameras. Figure 8 shows noise distribution measured for three different camera types capturing an object at the same position in the scene (Z = 1200 mm). Figure 9 shows the noise for the three different types of cameras across different depths. The ZED camera shows the highest noise values and the KinectV2 the lowest, across all depths. Both figures emphasize the difference in pattern noise across different camera types. This distinction allows to determine the source camera from the sensor noise of a depth image, as shown in [32]. To test this, the noise data collected with the three camera types (Kinect V1, Kinect V2, ZED) as described in Section 3, was used as the training model. Testing data was collected using new recordings of these cameras as well as additional camera units of the same type (Kinect V1, Kinect V2) with objects placed in the scene at positions similar to the training data. Additional testing data was obtained using Kinect V1 depth video sequences collected from a public database [36] and Kinect V2 sequences from private home recordings. A total of 300 test patches (each of 300 frames) were extracted from each of these test sequences and the noise statistics (histogram of noise magnitude, mean and variance) was calculated for each patch. Test samples totaled over 2100 examples. K-Nearest Neighbor was used to predict the source camera of the test data. Accuracy of the predictions are shown in Table 2. It can be seen that high accuracy rates were obtained for the Kinect V2 cameras; however the Kinect V1 cameras showed lower accuracy rates. This is not surprising due to the high level of noise in the latter. It can also be seen that higher accuracy rates were found for the data collected in the lab settings compared to those from private recordings and from the public database. It is interesting to note that the Kinect V2 cameras were detected even for units that were not in the training data, implying that the noise characteristics of the Kinect V2 are inherent to the camera type and do not significantly differ across units.

## 5. Distinguishing Real from Fake Faces Using Sensor Noise

We show that noise in depth images combined with a binary classification model can be used as an anti-spoofing tool for face recognition systems. For this purpose we used the 3D Mask Attack Data-set [37] which contains 17 real face sessions and 17 mask sessions (Figure 10). Each session consists of five videos of 300 frames each (both RGB and depth, of which we used only the depth frames). As a pre-processing step, we cropped a patch of size 98 × 98 around the face in all videos. Each video was segmented into 30 segments of 10 frames each. For each segment, we generated a noise image by evaluating the variance of depth values at each pixel across the 10 consecutive depth images (Equation (Equation 1)). The generated noise images were used as input feature vectors in training the binary classification model with labels corresponding to real/fake faces. Figure 11 shows the noise histogram for a real face (left column) and a masked face (right column) taken at 4 different regions on the face: (top to bottom); forehead, nose, chin and cheek. There is a clear difference in noise pattern between real and masked face. This is most likely due to the difference in material and temperature between skin and mask.

The training set consisted of 14 real sessions, and 14 mask sessions resulting in a total of 4200 noise images. The test set consisted of the remaining subjects not seen in the training and included 3 real and 3 fake sessions totaling in 900 noise images. Using a coarse Gaussian SVM (kernel scale 390, 5-fold cross validation), we achieved an accuracy of 95.11% success rate in determining real from fake on the test set. We were able to increases the accuracy to 96.67% by voting on the 30 noise images associated with each video and determine real from mask based on the majority vote. This high rate of success is most likely due to the difference in material and reflectivity between skin and mask which induce different noise characteristics (see Section 2.2). Determining real from masked faces can be exploited in detecting spoofing attacks [38,39,40].

## 6. Determining Scene Illumination Direction from Sensor Noise

As mentioned in Section 2.2, depth cameras do not perform well under outdoor lighting mainly due to the fact that natural light contains IR components which interfere with the IR light used by depth cameras [5,24]. Figure 12 shows this effect. A subject’s face was captured using a Kinect V2 camera with sunlight source on the right. The RGB image is shown on the left and the noise magnitude of the captured depth sequence (variance per pixel across all frames, as defined in Section 2.2) is shown on the right. It can be seen that noise values are larger on the right side of the face corresponding to a higher concentration of impinging light.

The effect of illumination on noise in depth images can be exploited to determine the source direction of the sunlight (relative to the object). We captured several depth sequences using a Kinect V2 camera, of a doll under sunlight. A sequence of 100 frames was captured every hour from 6 am to 6 pm and the noise image was computed for each sequence. Figure 13 shows the noise images for every hour from 6 am (top-left) to 6 pm (bottom-right). The region with largest noise values can be seen shifting from the right side of the doll to the left side corresponding to the change in sun position (see side, neck and nose of the doll). Figure 14 displays the noise ratio between the right and the left side of the doll’s face. For example, at 7 am the noise ratio was 2.1, i.e., the noise on the right was much greater than on the left, indicating that sun light direction was from the right. At 13 pm, the ratio was close to 1, i.e., equally distributed on both sides of the face, in accord with the sun light direction from above. Additionally, the strong sun light combined with the reflective surface (the plastic material of the doll), produced very strong reflections, inducing a large increase in noise, which caused saturation of the camera’s IR sensors and ultimately resulted in burned out pixels. This can be seen in Figure 13 where missing pixels at the top of the doll’s head can be seen shifting position with the sun’s motion.

This characteristic of changing noise ratios between left and right sides can be exploited for detecting inconsistencies—and hence detecting tampering—in depth images. Consider the example shown in Figure 15. Three objects are positioned on a table in the scene. The noise image of the depth sequence of this scene is shown in Figure 15. For each object, we examined the noise ratio between its right and left sides. Ratios are 0.6, 1.4 and 0.5 for the three objects, respectively, from left to right. The ratios indicate that two objects share the same lighting direction while the middle object has a different lighting direction. This inconsistency of lighting direction in the scene indicates some form of image manipulation.

## 7. Shadows in Depth Images

Depth sensing systems are based on two components: two RGB sensors in stereo cameras, and an IR projector and an IR sensor in structured light and ToF cameras. Shadows in depth images are portions of the scene that are observed by one component while blocked from view of the second component (Figure 16a). This typically occurs at the edges of objects in the scene. With ToF cameras, shadows form in the resulting depth image when the IR sensor sees a portion of the scene that is blocked from view of the IR projector. Depth cameras typically report shadows in the output depth images as pixels with 0 depth (visualized as black pixels in depth images). Shadow size is shown to be a function of object distance from the sensor (Figure 17), background distance from the sensor (Figure 18) and sensor configuration.

### 7.1. Shadow Size

Consider the scene and camera configuration shown in Figure 16a viewed as a projection onto the x-z plane of the camera coordinate system. The scene contains an object (blue box) with a background object or plane behind it. Denote by *b* the camera baseline, i.e., the distance between the two camera components. Let $d_{o}$ be the distance from camera sensor to the object and $d_{b}$ be the distance from camera sensor to the background. Let *s* denote the shadow width. (we consider only the horizontal geometry along the x-axis). Using simple geometry and triangle similarity, we have:(3)$$s=\frac{b}{d_{o}}\cdot \left(d_{b}-d_{o}\right)$$

It can be seen that the shadow width is independent of the horizontal positioning of the object. Thus, shadow width remains the same when the object is moved horizontally in the scene. However, shadow width changes when the object changes position in depth. Figure 16b shows two objects at depth positions $d_{o1}$ and $d_{o2}$. Using the relation in Equation (Equation 3) and equating for the baseline *b*, we obtain the relation between shadow widths s1 and s2:(4)$$s_{2}=\frac{d_{o1}}{d_{o2}}\cdot \frac{\left(d_{b}-d_{o2}\right)}{\left(d_{b}-d_{o1}\right)}\cdot s_{1}$$

The camera output is a 2D projection of the scene onto an image where each pixel value denotes depth. We consider the size of the object’s projection and that of its shadow. Using a standard camera model where the object’s projected size is inversely related to its distance $d_{o}$ from the camera sensor, we have that the projected sizes w1 and w2 of an object placed at distance $d_{o1}$ and $d_{o2}$ from the camera are related by:(5)$$w_{2}=\frac{d_{o1}}{d_{o2}}\cdot w_{1}$$

Finally, we map world units to image pixel units to allow measurements in the camera output image. Let α be the horizontal field of view angle of the camera, $N_{x}$ be the horizontal resolution of the image (the number of pixels per row of the camera sensor), and *d* be a distance from the camera sensor in millimeters (Figure 16c). The conversion between millimeters and pixels at depth *d* is given by:(6)$$N_{x}\;\left[pixel\right]=2d\cdot tan\left(\frac{\alpha }{2}\right)\;\left[millimeter\right]\;$$

To verify these calculations, we used the NYU Depth Dataset V2 dataset [41] which contains more than a thousand indoor scenes captured using a Kinect depth camera. Shadow widths were measured in each image and compared with the expected value calculated from Equations (Equation 3) and (Equation 6) using the Kinect V1 camera parameters. The average absolute difference between real and estimated shadow widths over all examples was μ=1.97 pixels with variance of $\sigma^{2}=1.4$. Implying consistency of the computed shadow width values.

### 7.2. Sensor Shadow as a Source Camera Identifier

Sensor shadow size can also be used as a source camera identifier since cameras differ in build, specifically in their baselines and fields of view which in turn, affect shadow size. Furthermore, the same camera with multiple resolution modes will generate different shadow sizes for the same scene, allowing the detection of the specific camera resolution. Figure 19 shows the same scene captured by three structured light cameras: Intel D415, Intel D435 and Microsoft Kinect. Camera parameters are: {b=55 mm, $\alpha =70{}^{\circ},N_{x}=1280$}, {b=50 mm, $\alpha =90{}^{\circ},N_{x}=1280$} and {b=75 mm, $\alpha =57.8{}^{\circ},N_{x}=320$}, respectively. Shadow width are correspondingly: 13, 8 and 3 pixels.

As an example of the use of shadow size as an indicator for source camera identification, we collected a set of depth images using the three cameras with the object at different depths. For each image, the measured shadow width was compared with the three possible shadow widths calculated using each of the cameras’ parameters. Table 3 shows the results. Each row represents an example, and lists the object distance to the camera $d_{o}$, the background distance to the camera $d_{b}$, the measured shadow width (in pixels) and the calculated shadow width (in pixels) for each of the three possible camera parameters. The true source camera is marked in gray. As can be seen, the calculated value closest to the measured shadow width indeed indicates the correct source camera.

### 7.3. Sensor Shadow Inconsistencies

Shadow size and its relation with object distance from camera can be exploited to detect suspected image manipulation or tampering. Tampering is suspected when the measured shadow width or position in the image does not match the theoretically expected value. In the following we describe several principles of this type of tampering detection.

For most depth sensing cameras, the IR projector is positioned to the right of the IR sensor, thus object shadows are formed primarily on the left edges of objects. Regardless, object shadows appearing on opposite sides of objects within the same scene, indicate inconsistency that may imply the image has been tampered with.

In depth images, tampering may induce scaling of the depth values of the object pixels. In this type of forgery the object and the shadow size do not change. However, the scene parameters $d_{o}$,$d_{b}$ change, thus affecting the expected shadow size. Figure 20 shows an example. The original image (Figure 20a) shows an object at distance 900 mm from the camera with the background at distance 1340 mm. Shadow width is measured as 12 pixels where the calculated width, given the camera parameters is 11.7 pixels. Figure 20b shows the forged image where depth values of the object were scaled to 500 mm. In this case the shadow width remains 12 pixels whereas the calculated width is 40.11 pixels, clearly, implying suspected tampering in the image.

However, even when corrected for object size, shadow width is still inconsistent. Figure 20c shows the object with depth values scaled to 500 mm and the object itself, together with it’s shadow, scaled to a size consistent with its new depth, using Equation (Equation 5). Shadow size in this example is thus scaled to 25 pixels which is still inconsistent with the calculated width of 40.11 pixels and forgery is detected here as well. This inconsistency in shadow width even with object re-sizing can be derived directly from Equations (Equation 3)–(Equation 5).

Finally, if the object and shadow are re-sized to obtain the shadow width as computed for the scaled depth value (40.11 mm), as shown in Figure 20d, we find inconsistency in the object’s size compared to its real world size. In the example, reverse-projecting the head in the image (from pixels to millimeters using Equation (Equation 6)) results in a head size of 230 mm (while real human heads average approximately ∼140 mm ).

## 8. Conclusions

This paper presented a study on noise in depth sensors. We showed how various parameters and scene characteristics can be deduced from the noise patterns in depth images. The varying pattern of noise across depths and horizontal positions in the scene, enabled determining the position and distance of objects from the camera using the noise in the image alone. Sensor noise can also determine the source camera type, and even the specific camera unit that was used. Noise patterns allowed distinguishing between real and masked faces with high accuracy. This can probably be attributed to the difference in material between skin and mask. Future studies can attempt to construct different noise models for different materials, thus, enabling detection of inconsistencies in noise statistics in scenes more comprehensively.

Distribution of sensor noise on objects in the scene can indicate the light source direction. Inconsistencies of these distributions across objects in the scene may indicate that the image was tapered with.

Finally we showed that missing depth data (shadows) in the image impose a relationship between object distance from background, object distance from camera and object size. Thus, manipulation of object depth, or manipulation of object size can be detected through shadow size measurements. Additionally, due to differences in camera parameters, shadow size can be used to determine source camera.

Extraction of scene and object characteristics from sensor noise, can be exploited in various applications such as determining the time of day, dealing with spoofing and determining whether an image has been forged.

The capabilities and methods presented in this study are specific for depth cameras and rely on the spatio-temporal noise patterns as well as image artifacts (such as depth shadows) that are unique to depth cameras.

## Figures and Tables

**Figure 1 sensors-22-05448-f001:**
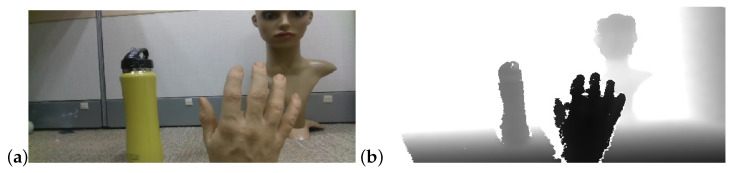
(**a**) RGB image. (**b**) Depth image—darker pixels indicate distances closer to the camera. (Intel RealSense D435).

**Figure 2 sensors-22-05448-f002:**
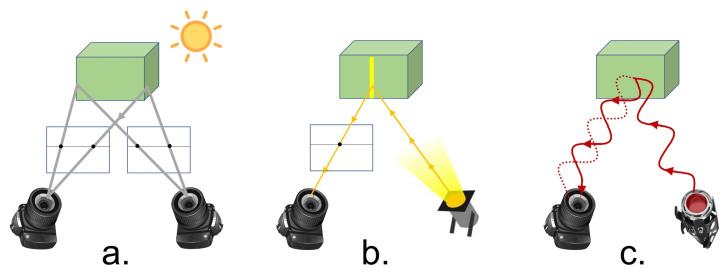
Depth Sensing technologies. (**a**) Passive stereo. (**b**) Structured light. (**c**) Time of Flight.

**Figure 3 sensors-22-05448-f003:**
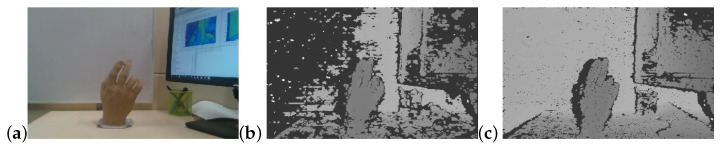
Noise is dependent on camera laser power. (**a**) RGB. (**b**) Low laser power. (**c**) High laser power.

**Figure 4 sensors-22-05448-f004:**
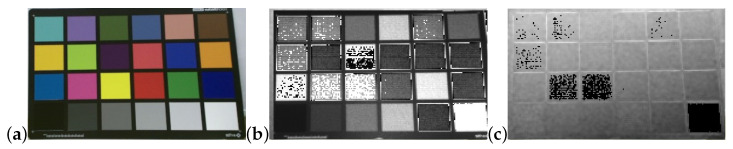
The effect of color in Kinect V2 camera. (**a**) RGB image. (**b**) IR image. (**c**) Depth image.

**Figure 5 sensors-22-05448-f005:**
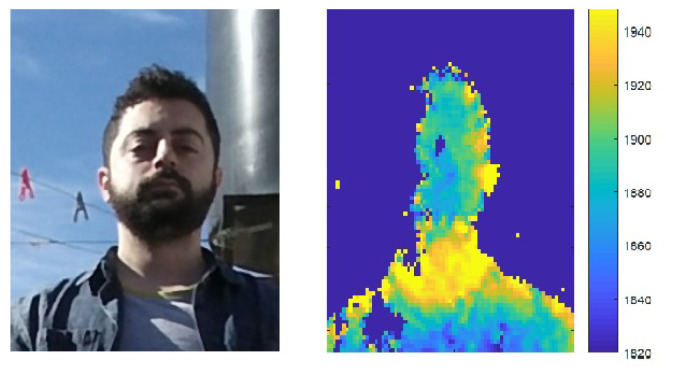
Strong sun light results in burned out pixels on the left side of the face.

**Figure 6 sensors-22-05448-f006:**
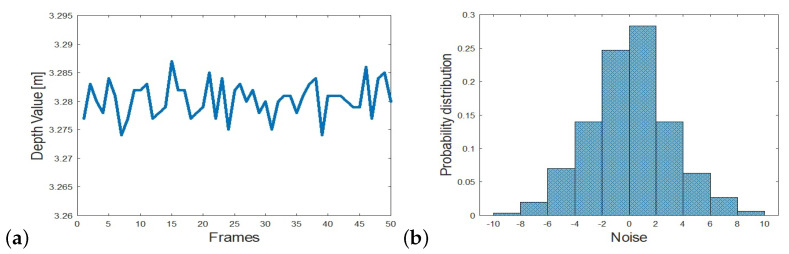
(**a**) Depth response at a pixel acquired over 50 frames by a KinectV2 [31]. (**b**) Histogram of noise at the pixel. *Noise magnitude* is defined as the variance of depth values and *noise variance* is defined as the variance of the absolute deviation values. In this example, noise magnitude is μ=0.094 cm, and noise variance is $\sigma^{2}=0.036$ cm.

**Figure 7 sensors-22-05448-f007:**
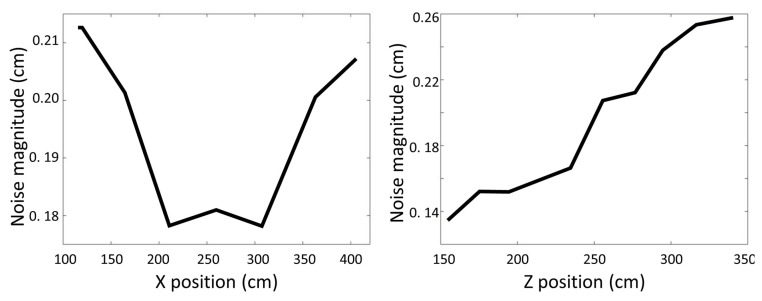
Noise magnitude as a function of horizontal position (X pos) (**left**) and depth (Z-pos) (**right**). Noise increases with horizontal distance from center and increases with depth (distance from camera).

**Figure 8 sensors-22-05448-f008:**
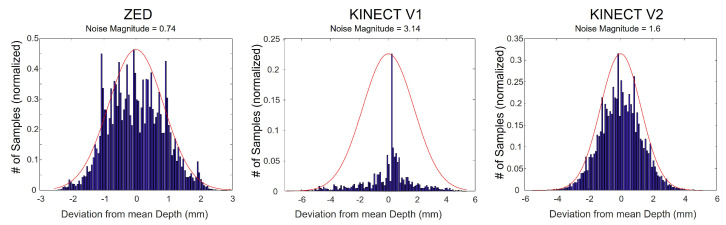
Histograms of deviations from the average depth value of an object at distance 1200 mm from the camera as captured by three types of cameras. Noise magnitude of these examples are (left to right) ZED 0.74, KinectV1 3.14 and KinectV2 1.6 (mm). The noise pattern differ between cameras. (Note the difference in scale of each plot).

**Figure 9 sensors-22-05448-f009:**
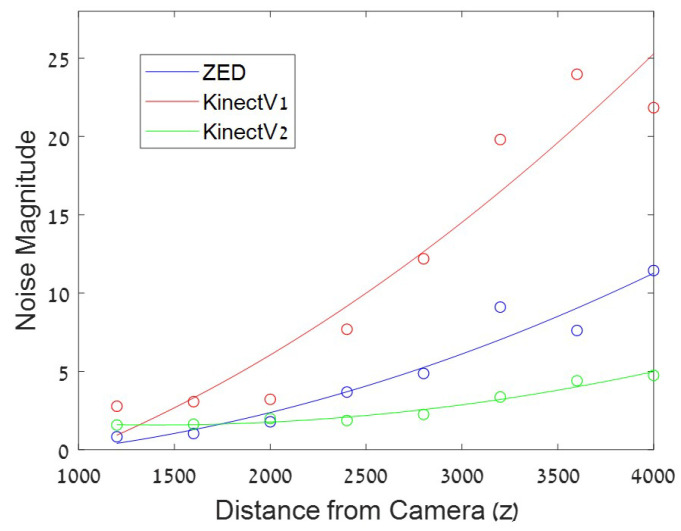
Camera noise at different distances from camera for three different camera types. Noise pattern differs significantly between cameras.

**Figure 10 sensors-22-05448-f010:**
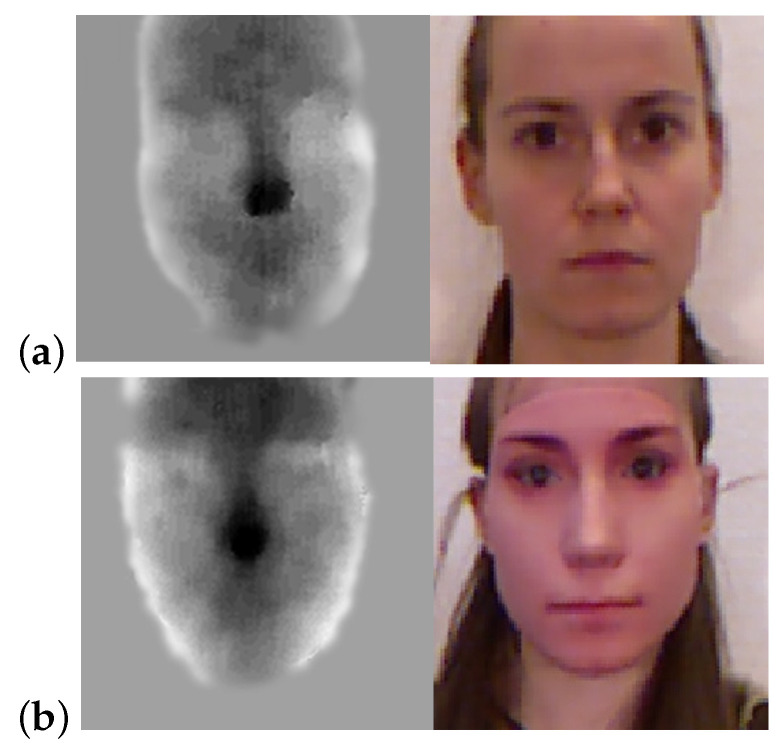
Examples from the 3D Mask Attack Dataset [37]. Depth image (**left**) and RGB image (**right**) of (**a**) real face and (**b**) masked face.

**Figure 11 sensors-22-05448-f011:**
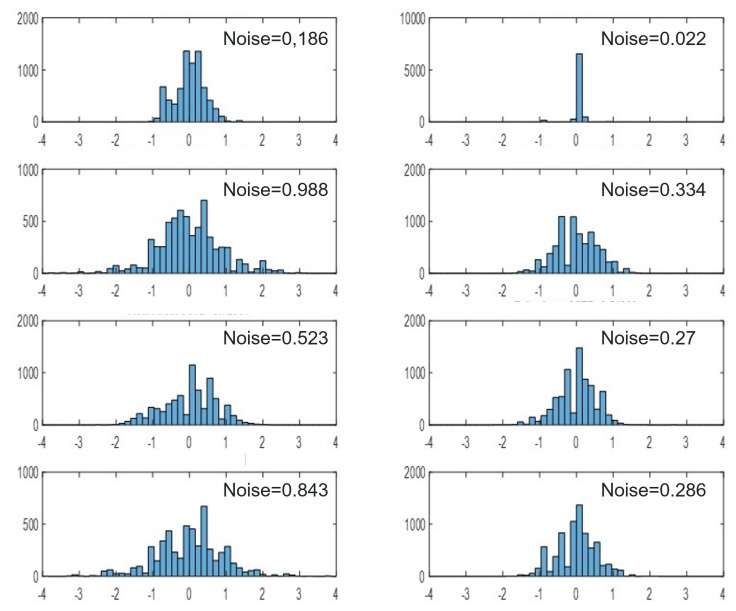
Noise histogram for a real face (**left column**) and a masked face (**right column**) taken at 4 different regions on the face: (**top** to **bottom**); forehead, nose, chin and cheek. Note the difference in noise pattern between the real and masked faces.

**Figure 12 sensors-22-05448-f012:**
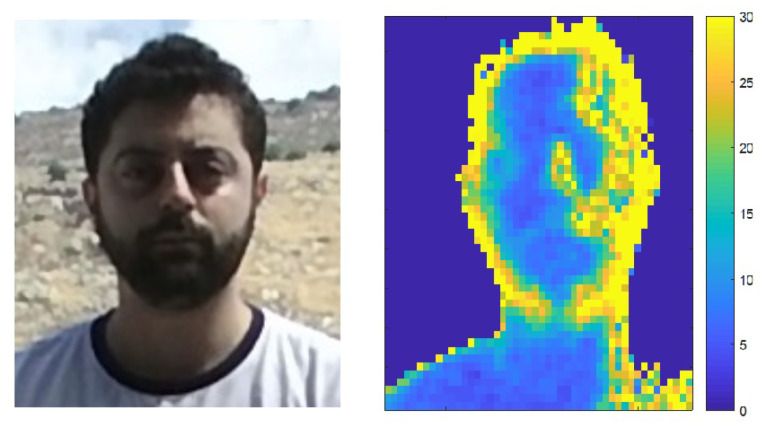
RGB image of a subject in sunlight (**left**), and corresponding depth noise image (**right**). Greater noise can be seen on the right side of the face indicating the sun’s direction.

**Figure 13 sensors-22-05448-f013:**
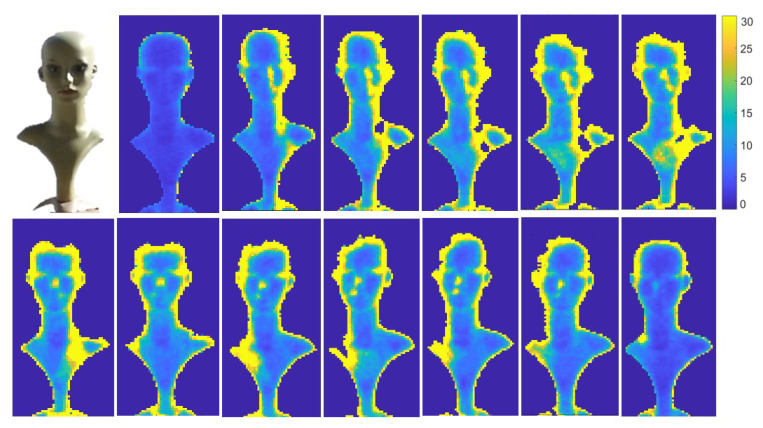
RGB image of a doll (**top left**) and noise images of the doll sequence captured by a KinectV2 camera, under sunlight every hour from 6am (**top left**) to 6pm (**bottom right**).

**Figure 14 sensors-22-05448-f014:**
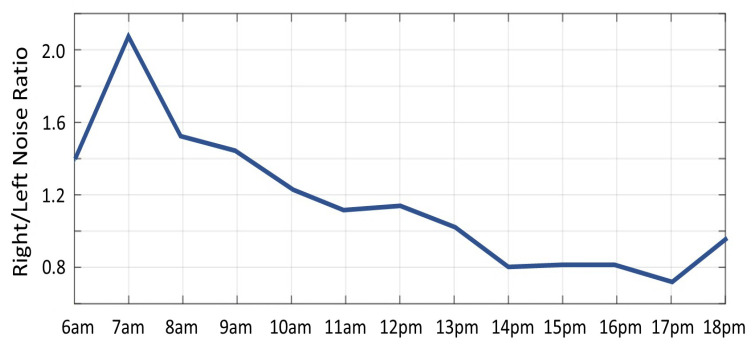
The noise ratio between right and left sides of the face plotted per hour.

**Figure 15 sensors-22-05448-f015:**
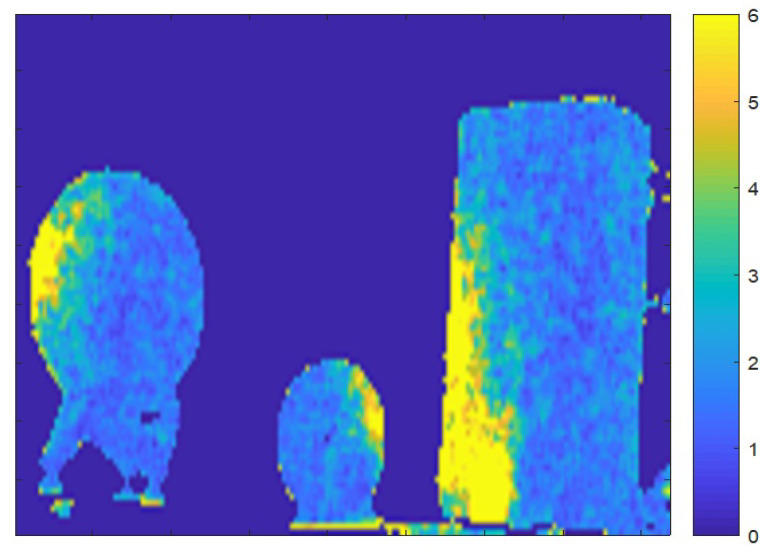
Example of detecting inconsistency in a scene based on scene illumination. A scene with 3 objects is captured using a depth camera. The noise image of the scene shows larger noise values on the right of the middle object and on the left of the other two objects.

**Figure 16 sensors-22-05448-f016:**
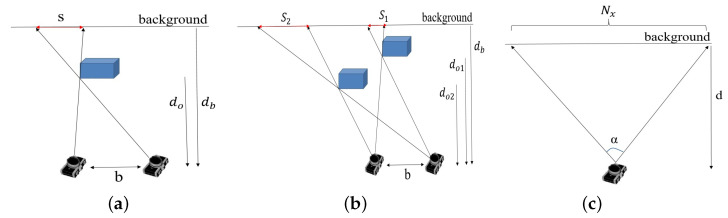
(**a**) A shadow in a depth image is the area observed by the left camera component (e.g., IR sensor) while blocked from view of the right component (e.g., IR projector). (**b**) Shadow size (S1,S2) depends on object distance from camera. (**c**) Conversion from world units to image units.

**Figure 17 sensors-22-05448-f017:**
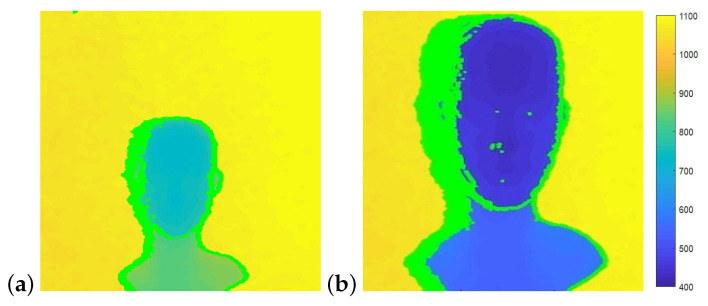
In both images, the distance from camera to the background is 1150 mm. The distance from camera to object is (**a**) 770 mm and (**b**) 450 mm. Shadow pixels are marked in green on the left side of the head. The smaller the distance from object to camera, the wider the shadow.

**Figure 18 sensors-22-05448-f018:**
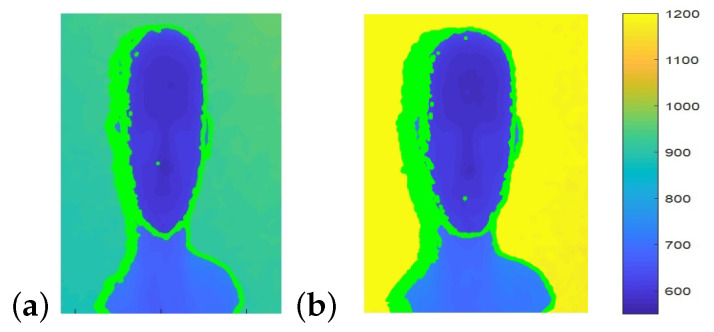
In both images, the object’s distance from camera is 600 mm. The distance of the background from the camera is (**a**) 880 mm and (**b**) 1200 mm. Shadow pixels are marked in green on the left side of the head. The greater the distance between object and background, the wider the shadow.

**Figure 19 sensors-22-05448-f019:**
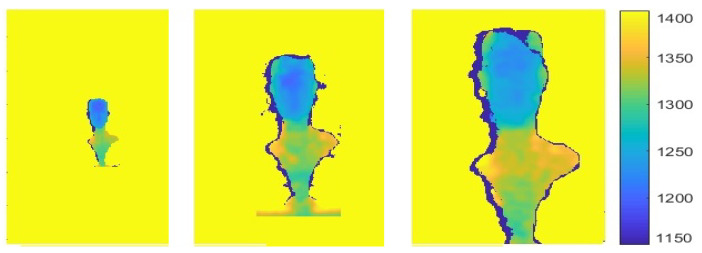
The same object captured by three different depth cameras positioned at the same distance from the camera. Shadow size in pixels (left to right): Kinect V1 (3), Intel D435 (8) and Intel D415 (13).

**Figure 20 sensors-22-05448-f020:**
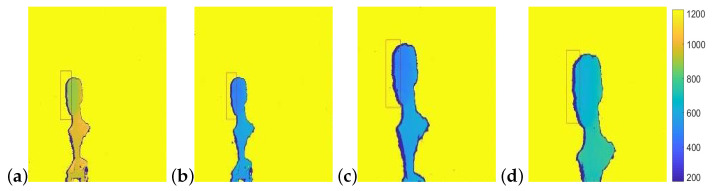
Manipulating depth images. (**a**) Original depth image. Object and background distances from camera are 900 mm, and 1340 mm. Shadow width is calculated as 11.675 pixels and measured 12 pixels. (**b**) Object depth values have been scaled to 500 mm. Shadow width is now calculated as 40.11 pixels with shadow width in image, remaining 12 pixels. (**c**) Object depth values have been scaled to 500 mm and the object re-sized consistently. The shadow width is now calculated as 40.11 pixels and measured width is 25 pixels. (**d**) Depth values have been scaled to 500 mm and the object re-sized to correct for the shadow width. Shadow width now measures 40 pixels in the image. The head, however, now corresponds to a head size of 230 mm, which is much larger than a human head.

**Table 1 sensors-22-05448-t001:** Depth (Z-value) and horizontal position (X-value) prediction from noise.

	Correct X-Z Prediction	Avg Z Error	Avg X Error
1-rank	73%	9.61 cm	38.54 cm
2-rank	92%	1.54 cm	6.62 cm
3-rank	97%	0.31 cm	1.84 cm

**Table 2 sensors-22-05448-t002:** Camera source identification results.

Camera Unit	% Correct Camera Type Identification
KinectV2 (unit #1) (Training)	98%
KinectV2 (unit #2)	92%
KinectV2 (unit #3)	95%
KinectV2 (private Cam1)	95%
KinectV2 (private Cam2)	87%
KinectV2 (private Cam3)	86%
KinectV2 (private Cam4)	97%
KinectV1 (unit #1) (Training)	90%
KinectV1 (unit #2)	74%
KinectV1 (unit #3)	75%
KinectV1 [36]	92%
KinectV1 [36]	65%
KinectV1 [36]	68%
ZED (training)	96%

**Table 3 sensors-22-05448-t003:** Camera source identification. The measured shadow width is compared with the three expected shadow sizes calculated using the parameters of each of the three cameras. The true source camera is marked in gray.

$d_{o}$	$d_{b}$	Measured	D435	D415	KinectV1
330	930	63	63	98	85
450	1050	42	41	63	55
750	1350	20	19	30	26
940	1410	10	19	29	7
1125	1810	5	11	17	7
600	1100	36	24	38	16
750	1030	18	11	18	7
750	1350	29	18	30	12

## Data Availability

The publicly available 3D Mask Attack Data-set (3DMAD) [37] was used in this study.

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
