# Peer review of "What Can We Learn from Depth Camera Sensor Noise?"

_sensors, 2022, doi:10.3390/s22145448_

Round 1

Reviewer 1 Report

Interesting topic and excellent work. It is found that the several types of information can be deduced from the noise of the depth camera sensor, including object depth, object location, the source camera type. Also, distribution of sensor noise on objects in the scene can indicate the light source direction and differences in noise pattern can distinguish between real and masked individuals. What is more, missing depth data in the scene impose a relationship between object distance from background, object distance from camera and object size. The paper is well written.

Comments:

The average X error seems large in table 1, why that?

Reviewer 2 Report

The paper deal with using of camera sensor noise to receive some information about the object. Introduction sufficiently describes the problem and theoretical background guide the reader through several depth sensing technologies. Camera sensor noise is defined in minimal matter in chapter 2.2.

In chapter 3 three cameras was used to show possibility to find the position of the object from sensor noise. For better understand the results, more details about SVM classifier (structure of the model, training data, type of learning algorithm, etc.) will be good. Also the comparison from standard usage of depth cameras will be an advantage. An information about metrology uncertainty of the object position will be an advantage.

The determination of masked faces need more information about the process of classification and the results should be used on different condition (not only presented 3D Mask attack data-set).

The same apply for illumination and shadow. Also more investigation about influences of emissivity (reflectivity) of used materials (with their direction and temperature dependence) will help to show proposed methodology.
